American Society for Microbiology | Microbiology Spectrum

# Unlocking genome engineering in *Alcaligenes faecalis* by exploiting its native type I-F CRISPR-Cas

Wanting Cheng,[1] Jiaxin Li,[1] Lei Lei,[2] Yuxuan Zhu,[1] Siqi Luo,[1] Xueqing Wang,[1] Qinghui Zhang,[1] Miaomiao Cao,[1] Yanli Zheng,[2] Wenfang Peng[1]

**ABSTRACT** *Alcaligenes faecalis* is an environmentally significant bacterium for pollutant biodegradation and aerobic denitrification, yet its genetic engineering has been hindered by a lack of high-throughput tools. Conventional methods like homologous recombination are time-consuming and cannot achieve large genomic deletions, while technologies based on heterologous CRISPR-Cas systems failed due to cytotoxicity. This study resolves these limitations by developing a genome editing toolkit based on the endogenous type I-F CRISPR-Cas of *A. faecalis* J481. The toolkit enables efficient single-gene knockout and accomplishes the previously unattainable precise deletion of large genomic fragments. By engineering a *PheS*-mutant counterselection marker, we achieved rapid plasmid curing, allowing two rounds of large-fragment removal (~47 kb total) within 5 days. This breakthrough provides the first CRISPR-based platform for complex genome engineering in *A. faecalis*, overcoming intrinsic constraints of heterologous systems. The work establishes a scalable genetic toolbox to enhance *A. faecalis'* capabilities in bioremediation and eutrophication control. Moreover, the strategy of harnessing endogenous CRISPR-Cas systems offers a blueprint for developing advanced genome editing tools in other prokaryotes.

**IMPORTANCE** This study breaks through the longstanding genetic engineering bottleneck in an environmentally crucial bacterium, *Alcaligenes faecalis*, by creating a fast, efficient, and versatile toolkit using its native CRISPR-Cas system. This enables complex edits, such as large genomic deletions previously impossible, unlocking new potential for bioremediation and eutrophication control, providing a blueprint for other prokaryotes, and setting a precedent for genetic tool development in other hard-to-engineer microbes.

**KEYWORDS** *Alcaligenes faecalis*, endogenous CRISPR-based genome editing, large genomic deletion, 4CP/PheSv counterselection, prokaryotic engineering

*A*lcaligenes faecalis is a gram-negative bacterium commonly found in the environment. With attractive physiological features in biodegradation, *A. faecalis* strains have been widely applied as useful tools for addressing environmental issues. For instance, biodegradation using *A. faecalis* has been one of the environmentally friendly approaches for efficient degradation of a broad range of toxic chemicals, such as ochratoxin A (1, 2) and quinolinic acid (3), as well as discarded polymers (4–7). Interestingly, it was reported that the *A. faecalis*-performed ochratoxin A degradation could allow for alleviating the immune injury and inflammation induced by the toxin (2). In addition, as an aerobic denitrifying bacterium, *A. faecalis* has been also extensively studied and broadly applied in eutrophication due to its superior nitrogen removal capacity compared to autotrophs (8–11). While these excellent applications have drawn a great deal of interest in the strains of *A. faecalis*, efficient genetic manipulation methods

**Peer Reviewer** Zhi-Qiang Xiong, University of Shanghai for Science and Technology, Shanghai, China

Address correspondence to Wenfang Peng, wenfang@hubu.edu.cn, or Yanli Zheng, Yanli.Zheng@whpu.edu.cn.

Wanting Cheng and Jiaxin Li contributed equally to this article. Author order was determined alphabetically.

The authors declare no conflict of interest.

would help develop and further improve *A. faecalis* as an ideal chassis for biodegradation applications and eutrophication studies.

To date, genetic manipulations in *A. faecalis* have predominantly relied on conventional methodologies. While these approaches have facilitated relatively straightforward genetic modifications, such as single-gene knockout, they remain inadequate for implementing more sophisticated genome editing options. For example, the classical two-step homologous recombination method has been routinely used for gene knockout in *A. faecalis* (8, 12), which typically involves two steps: replacing the target gene with a selection cassette and subsequently removing the selection cassette through counterselection, generally requiring at least several weeks to obtain a pure mutant strain (13). Despite its utility, this approach becomes highly time-consuming when multiple genes are targeted for deletion and, in some cases, is nearly unattainable if the removal of large genomic fragments is required. Thus far, large genomic fragment deletion has not been attained by using conventional genetic manipulation methods in *A. faecalis* yet, indicative of its being highly challenging. Consequently, there is a pressing need for more efficient high-throughput genome engineering tools for *A. faecalis*.

Over the past decades, CRISPR-based technologies, primarily utilizing CRISPR-Cas9, have been extensively applied for versatile genetic engineering purposes across diverse organisms, including numerous bacterial species (14). However, reports on their applications in *A. faecalis* remain conspicuously absent from the current literature. We initially undertook multiple attempts to implement genome editing in *A. faecalis* using CRISPR-Cas9, but these efforts proved unsuccessful, as no transformants were obtained. This outcome suggests intrinsic cytotoxicity of the heterologous Cas nuclease to the host cells, consistent with previous observations in other bacteria (15). In recent years, several native type I CRISPR-Cas systems have been successfully exploited for genome editing in their corresponding hosts (16, 17). Specifically, type I-F CRISPR-Cas systems demonstrated remarkable efficacy in targeted genome editing in *Pseudomonas aeruginosa*, *Zymomonas mobilis*, and *Acinetobacter baumannii* (18–21). More recently, their application has been extended to genome engineering and gene repression in *Pseudomonas chlororaphis* (22). The feature of comparatively fewer components of type I-F systems among the type I subtypes renders them particularly amenable to heterologous genome editing in diverse microbial species. These advancements have unveiled novel prospects for repurposing *A. faecalis*-derived CRISPR-Cas as intrinsic genome editing tools. However, systematic investigations exploring native CRISPR-Cas systems for genome editing in *A. faecalis* are still notably absent.

In this study, we functionally characterized the native type I-F CRISPR-Cas of *A. faecalis* J481 and engineered it into a versatile genome editing platform. This platform facilitated the accomplishment of diverse genome editing tasks, including targeted single-gene knockout, *in situ* gene tagging, and precise removal of large genomic fragments. Furthermore, we developed a counterselection marker based on a PheS mutant, enabling efficient curing of genome editing plasmids. This advancement allowed us to complete two rounds of CRISPR-based genome editing within merely 5 days, achieving the removal of two large genomic regions encompassing approximately 47 kb in total. This work significantly expands the genetic manipulation repertoire for *A. faecalis*, offering valuable insights for leveraging endogenous CRISPR-Cas systems for strain development and metabolic engineering in other prokaryotes.

## RESULTS

### Functional characterization of the endogenous CRISPR-Cas of *A. faecalis* J481

Based on the publicly available genome sequence of *A. faecalis* J481 (GenBank acc. no. CP032521), a type I-F CRISPR-Cas system was identified, comprising a cluster of *cas* genes and two CRISPR arrays (Fig. 1A). The *cas* genes are present in the order of *cas1-cas3-cas8(csy1)-cas5(csy2)-cas7(csy3)-cas6(csy4)*, being architecturally the same as the previously well-studied type I-F systems (18, 23). They are organized as two operons with the *cas1* gene and the featuring type I-F *cas3* (actually a *cas2-3* fusion) forming

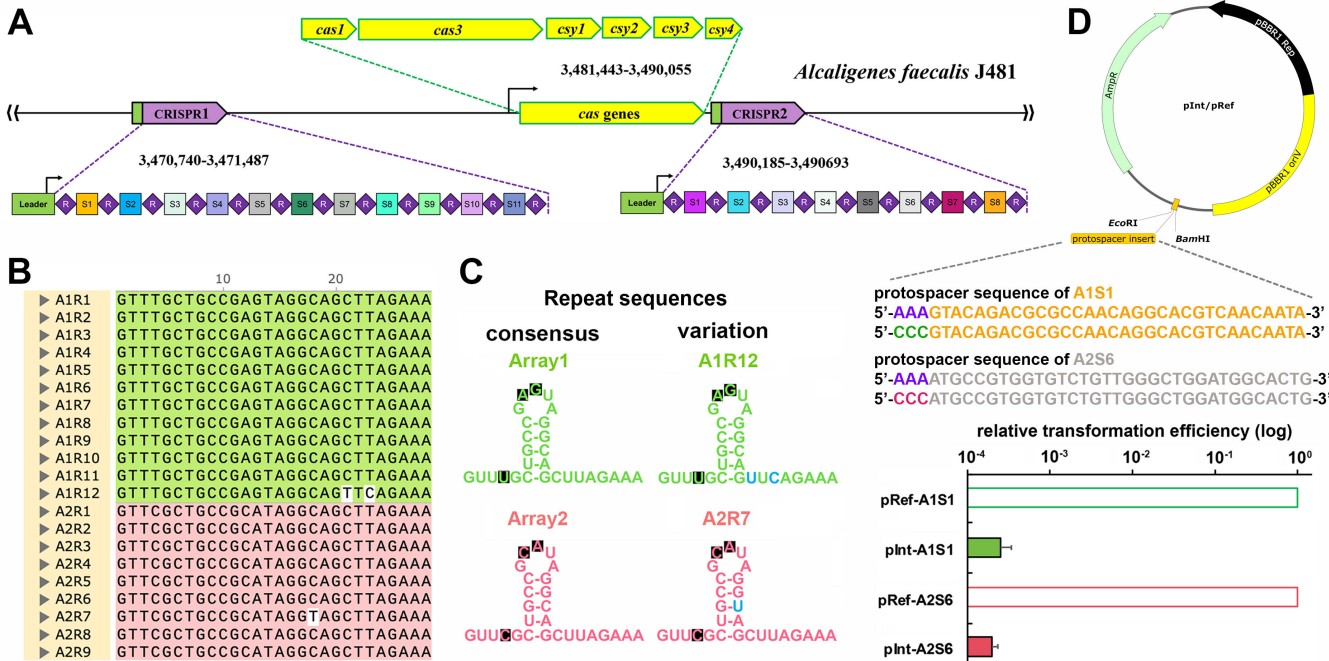

**FIG 1** Characterization of the endogenous type I-F CRISPR-Cas of *A. faecalis* J481. (A) Schematic of two CRISPR arrays and a *cas* locus in *A. faecalis* J481. (B) Representative and alignment of repeat sequences. Repeats from CRISPR1 and CRISPR2 are shaded in green and magenta, respectively. Palindromic sequences are indicated with arrows. (C) Predicted secondary structures of the repeats. Nucleotide variations are indicated. (D) Plasmid interference assay. Interference plasmids (pInt) contain a protospacer comprising a spacer preceded by a 5′-CCC-3′ PAM, while the corresponding reference plasmids (pRef) feature a 5′-AAA-3′ motif substituted for the 5′-CCC-3′ PAM. *Eco*RI and *Bam*HI restriction sites used for protospacer insertion are indicated. Nucleotide sequences of the protospacers for A1S1 and S2S6 are presented. Two interference plasmids, pInt-A1S1 and pInt-A2S6, were used to assay DNA interference activity of the type I-F CRISPR-Cas. Transformation efficiencies of each interference plasmid are expressed as relative values to the efficiencies of their corresponding reference plasmids (pRef-A1S1 and pRef-A2S6), the latter of which are set to 1.0. Experiments were done in triplicate. Error bars represent the standard deviation of the mean.

one, while the four *csy* genes forming the other. Two CRISPR arrays, Array1 and Array2, were found at two different locations, with Array1 being about 10 kb upstream of, and Array2 immediately downstream of, the *cas* locus. Both arrays are typically composed of 28-nt repeats separated by spacers in lengths of 32 or 33 nt. Within each array, the repeat sequences tend to be identical. In Array1, the sequences of most of the repeats are 5′-GTT<u>T</u>GCTGCCG<u>AG</u>TAGGCAGCTTAGAAA-3′ except for A1R12 harboring two nucleotide mutations; while in Array2, all but A2R7 have a consensus sequence of 5′-GTT<u>C</u>GCTGCCG<u>CA</u>TAGGCAGCTTAGAAA-3′. Between the sequences, there are three nucleotide differences, which are underlined (Fig. 1B). Interestingly, in A2R7, the C to U mutation turns the original G-C match into a G-U pair in the stem region, while the other nucleotide changes sit out of the stem region (Fig. 1C). Predictably, all these variations would have no detectable influence on recognition and processing of the CRISPR repeats by Cas6 (23), which was subsequently validated in the plasmid interference assay (Fig. 1D).

Given the reliance of CRISPR-based genome editing technologies on the DNA targeting efficacy of CRISPR-Cas systems, it is imperative to delineate whether the type I-F system of *A. faecalis* exhibits constitutive activity akin to that observed in some well-characterized systems (18, 24). In this regard, we assayed *in vivo* plasmid DNA cleavage by repurposing the DNA targeting functionality of the endogenous CRISPR-Cas. Two interference plasmids, pInt-A1S1 and pInt-A2S6, were constructed, with each carrying the spacer sequence (S1 or S6) of the respective CRISPR arrays. Both spacers were preceded by an empirically determined functional 5′-CCC-3′ PAM (18). For the corresponding reference plasmids, pRef-A1S1 and pRef-A2S6, the PAM was replaced with a motif identical to the last three nucleotides of the repeat, mimicking the genetic

structure of the genomic CRISPR array. Following the individual introduction of the plasmids into *A. faecalis* cells via electroporation, it was observed that the transformation rates associated with pInt-A1S1 or pInt-A2S6 were >3,700-fold lower than those with the reference plasmids (Fig. 1D), indicating that the interference plasmids were considered as invaders by the type I-F CRISPR-Cas, therein triggering potent defense responses. These results demonstrated that the type I-F CRISPR-Cas of *A. faecalis* is constitutively active under standard laboratory conditions, thereby presenting a viable candidate for genome editing applications.

## Development of a genome engineering toolkit for *A. faecalis* based on the endogenous CRISPR-Cas

After establishing the potent interference activity of the endogenous type I-F CRISPR-Cas system of *A. faecalis* against the plasmid invaders, we sought to leverage this system for genome editing by taking the *avs2* gene (*D6I95_03165*), which encodes a homolog of type 2 antiviral STAND proteins (25, 26), as an editing target. To this end, a genome editing plasmid for *avs2* knockout (pKO-*avs2*) was fabricated, harboring an artificial CRISPR array and a repair template. The artificial CRISPR contains a spacer identical to a 5′-CCC-3′ PAM-preceded 32-bp sequence (protospacer) of *avs2*, while the repair template consists of two homology arms, namely the approximately 500 bp DNA fragments up-flanking and down-flanking of *avs2*, respectively. It was experimentally determined that the 500 bp homology is sufficient for facilitating recombination in *A. faecalis* (Fig. S1). We envisioned that crRNAs generated from the artificial CRISPR would direct the CRISPR-Cas effector complex to the protospacer sequence in *avs2*, exerting a cut of the chromosome. Meanwhile, the repair template would facilitate homologous recombination to achieve the desired deletion, which can be identified using the primer set illustrated in Fig. 2A.

Following the introduction of the pKO-*avs2* plasmid (498.03 ng of DNA) into the wild-type J481 cells via electroporation, a total of 242 transformants formed colonies on an ampicillin selection plate, corresponding to a transformation efficiency of 485.91 CFU/µg plasmid DNA. Among them, 16 were randomly chosen for genotypic analysis through colony PCR. As depicted in Fig. 2B, four out of the tested colonies exhibited the expected deletion (Δ*avs2*) with a predicted size of 744 bp, which was subsequently validated by Sanger sequencing of the PCR products (Fig. 2C). Additionally, a plasmid expressing a portal protein (NCBI ref. no. NP_955536; encoded by the *E. coli* lambda phage *B* gene), pELB, could be easily introduced into the Δ*avs2* mutant but not into the wild-type J481 cells (Fig. 2D). This finding is consistent with a previous report indicating that Avs systems can be activated by various phage portal proteins to induce cell death (26), thus confirming the antiviral defense function of *A. faecalis* Avs2. The anti-phage function of the Δ*avs2* strain could be fully restored through carrying an Avs2-expressing plasmid, pAvs2 (Fig. 2D). Collectively, these results demonstrated the efficacy of the native type I-F CRISPR-Cas of *A. faecalis* for genome editing, establishing its value as a powerful toolkit for gene function characterization in *A. faecalis*.

## Optimization of the endogenous CRISPR-based platform to enable large fragment deletion

In addition to single-gene knockout, we endeavored to delete a ~30 kb genomic region using the genome editing plasmid pDel-30k. The DNA fragment includes 34 genes (*D6I95_00365* to *D6I95_00530*) and is recognized as a prophage by PhiSpy (27). Initially, despite numerous attempts, only one or two transformants could be obtained from each transformation, none of which exhibited the anticipated genetic modifications. Based on our previous experimental experiences (18, 28), we hypothesized that, for this more challenging editing approach, the efficiency of plasmid transformation would be crucial in ensuring successful editing outcomes.

Previous studies have indicated that enhanced editing efficiency can be achieved through the suppression of bacterial restriction-modification (R-M) systems (18, 29); this

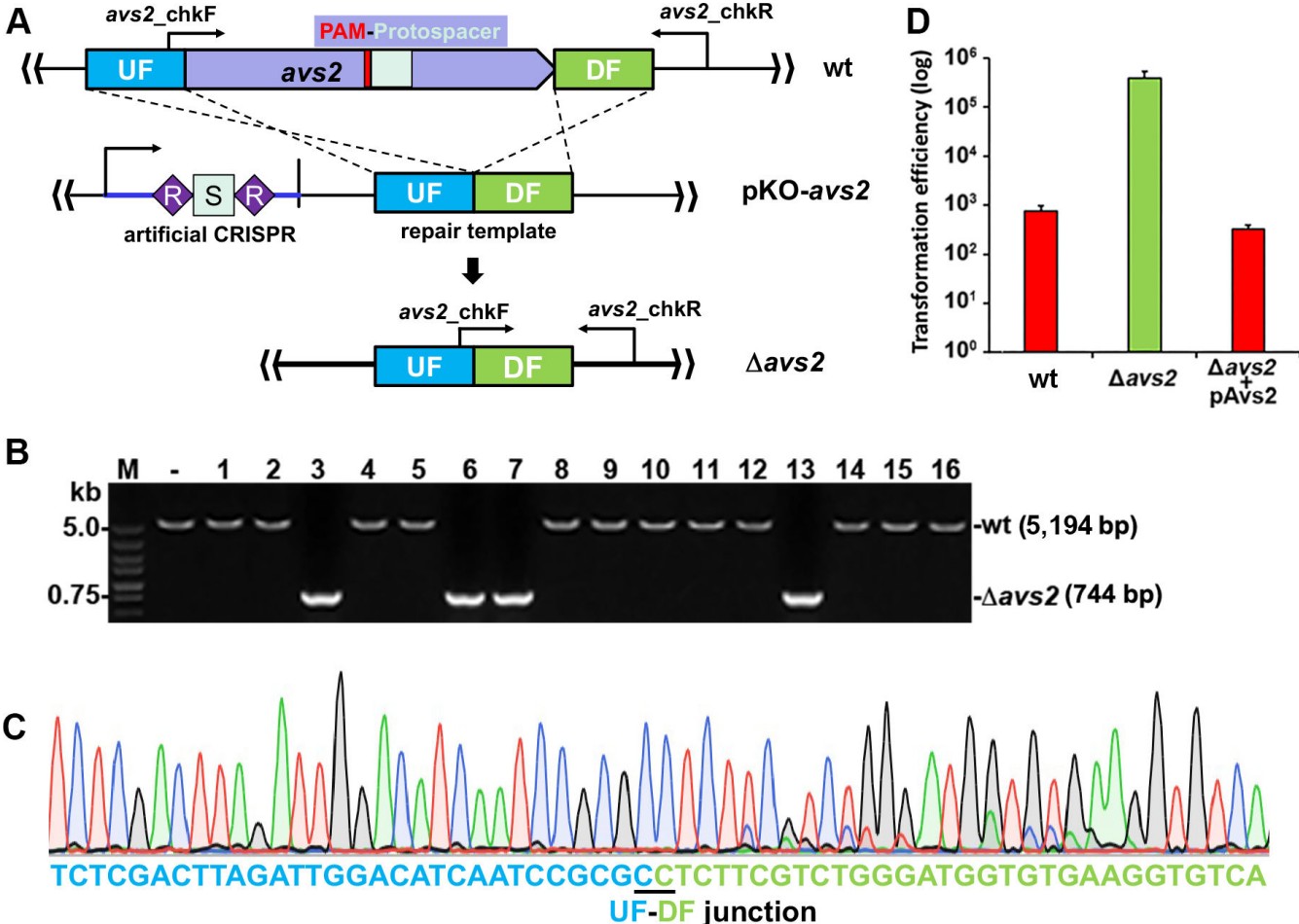

**FIG 2** Establishment of the endogenous CRISPR-based genome editing tool in *A. faecalis* J481. (A) Schematic showing the strategy of *avs2* knockout. An artificial CRISPR and a repair template consisting of up-flanking (UF) and down-flanking (DF) DNA fragments of *avs2* are included in an all-in-one plasmid. PAM and protospacer are shown. (B) Colony PCR screening of deletion mutants of *avs2* using a gene-specific primer set, *avs2*_chkF/*avs2*_chkR, as indicated in panel A. Predicted sizes of PCR products of wild-type (wt) strains and deletants (Δ*avs2*) are indicated. −, PCR amplification using genomic DNA of *A. faecalis* J481 as a DNA template; M, DNA size marker. (C) Representative chromatograph of Sanger sequencing result. UF-DF junction is indicated. (D) Plasmid interference assay demonstrating the function of *avs2* in antiviral defense. A plasmid-borne phage portal protein triggered an antiphage response in Avs2-expressing strains, including wt cells or Δ*avs2* carrying pAvs2 plasmid. Three replications were performed for each DNA sample. Error bars represent the standard deviation of the mean.

phenomenon is presumably applicable to various other bacteria, including *A. faecalis*. Three R-M systems were identified in *A. faecalis* J481 using DefenseFinder (30), including (i) a type I module encoding *hsdR1* (D6I95_14420), *hsdS* (D6I95_14425), *hsdR2* (D6I95_14430), and *hsdM* (D6I95_14435); (ii) a type III system comprising a restriction endonuclease (REase, D6I95_06790) and a methyltransferase (MTase, D6I95_06785); and (iii) a type IV *mrr* element (D6I95_00770). Transformation experiments were conducted using 10 ng of the *A. faecalis-E. coli* shuttle plasmid pAE1, which was independently prepared from *E. coli* DH5a ($dam^+dcm^+$) and Trans110 ($dam^-dcm^-$) strains. The transformation efficiency achieved with pAE1 prepared from DH5a cells was $(2.22 \pm 0.31) \times 10^5$ CFU/µg plasmid DNA, demonstrating a 76-fold reduction compared to that obtained with pAE1 extracted from Trans110 cells. This significant difference indicates that the IV R-M system, which has been shown to specifically target modified DNA in other bacterial species (31), plays a crucial role in foreign DNA restriction.

Consequently, we constructed a deletion mutant of the *mrr* (D6I95_00770) gene (Δ*mrr*). Notably, following transformation with the pAE1 plasmid prepared in *E. coli* DH5a, the Δ*mrr* mutant demonstrated a transformation efficiency of $(1.57 \pm 0.42) \times 10^7$ CFU/µg

plasmid DNA, representing an approximately 130-fold increase relative to the wild-type J481 strain, which exhibited a transformation efficiency of $(1.20 \pm 0.63) \times 10^5$ CFU/µg plasmid DNA. This Δ*mrr* strain was subsequently utilized to host the pKO-*avs*2 plasmid for reassessing genome editing efficiency. Remarkably, using an amount of 72.24 ng plasmid DNA, more than 500 transformants were observed, which is in sharp contrast to the yield of merely 1–2 transformants when hosted by the wild-type J481 cells. Fifteen Δ*avs2* mutants with the intended modifications were identified from 16 randomly selected transformants (Fig. 3A), demonstrating a considerably high editing efficiency of 93.75%.

Based on the findings, we proceeded with genome editing using the pDel-30k plasmid under the Δ*mrr* background. This time, a total of 21 transformants were obtained after electroporating 4,224.93 ng of pDel-30k into Δ*mrr* competent cells, all of which were genotypically analyzed by colony PCR. As shown in Fig. 3B, a band with an expected size of 840 bp could be amplified from 10 strains. According to the results of Sanger sequencing of the PCR products, these strains each harbored the desired ~30 kb deletion (Fig. 3C). Therefore, the Δ*mrr* strain enabled, for the first time in *A. faecalis*, genome editing involving large fragment deletion with an efficiency of 47.62%. Remarkably, using this optimized platform, highly efficient in-frame knockout of genes spanning various sizes was readily achieved, including a 774 bp deletion in *ectD* (*D6I95_05840*) with 100% efficiency (16/16), a 2,634 bp deletion in the *3Hp* operon (*D6I95_13695-D6I95_13705*) with 93.75% efficiency (15/16), and a 3,360 bp deletion in *cas3* (*D6I95_16065*) with 87.5% efficiency (14/16; Fig. S2). These results substantiate the broad applicability of the native CRISPR-based genome editing toolkit developed for *A. faecalis*.

## Implementation of the type I-F CRISPR-based tool for *in situ* gene tagging

We subsequently evaluated the efficacy of this type I-F CRISPR-based tool for performing *in situ* gene tagging in *A. faecalis*, thereby facilitating *in vivo* functional investigation of target genes. To this end, we designed and constructed the genome editing plasmid, pHT-*qatA*, for *in situ* His-tagging of the *qatA* gene (*D6I95_14445*), which encodes the NTPase component of the QatABCD phage defense system (32).

A critical challenge in this editing lies in the necessity to modify the sequence after tag insertion to render the protospacer unrecognizable by the CRISPR-Cas system, thus ensuring cell viability. Through careful examination of the sequences surrounding the *qatA* stop codon, we identified a 5′-CCC-3′ PAM located on the non-coding strand. Consequently, the 32-nt sequence immediately downstream of the PAM was considered a protospacer. A tag coding sequence (G4S-6xHis) was strategically designed to be inserted immediately upstream of the *qatA* stop codon, thereby interrupting the protospacer (Fig. 4A).

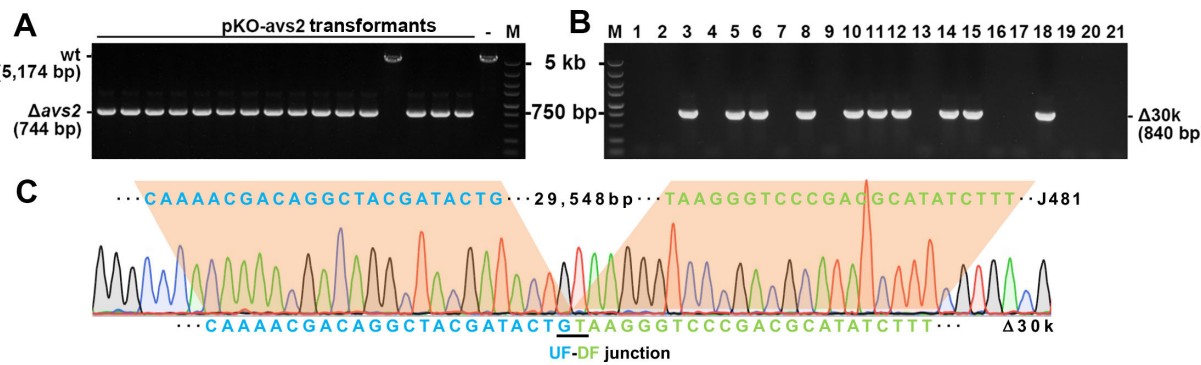

**FIG 3** Achievement of large fragment deletion using an optimized genome editing tool. (A) Colony PCR screening of *avs2* deletion mutants. (B) Colony PCR screening of 30 kb deletion mutants. (C) Representative chromatograph of Sanger sequencing result. The up-flanking and down-flanking (UF-DF) junction after the 29,548 bp deletion is indicated.

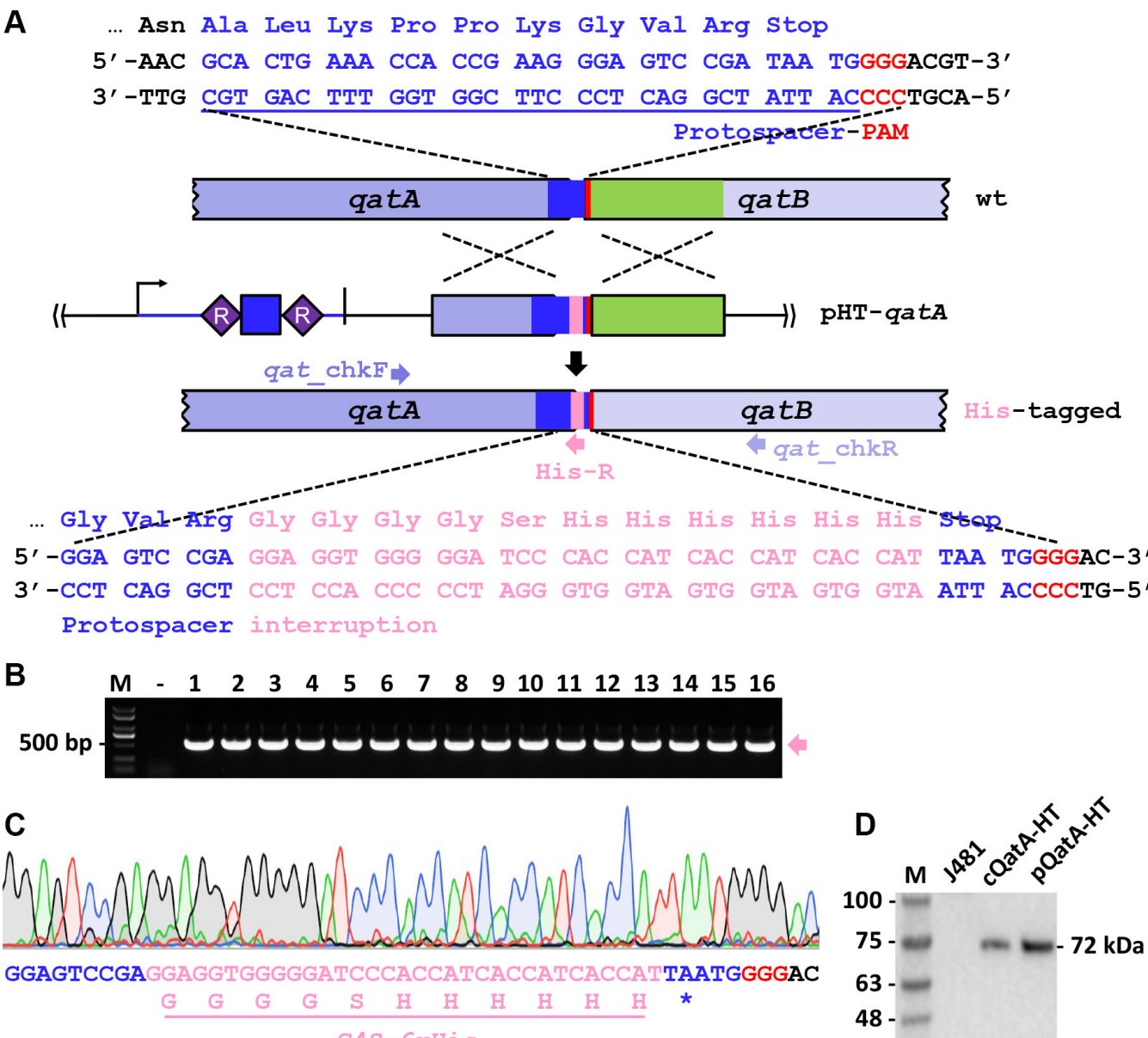

**FIG 4** Implementation of type I-F CRISPR-based tool for *in situ* protein tagging. (A) Schematic diagram illustrating the strategy for *in situ* His-tagging of QatA. The genome editing plasmid pHT-*qatA* expresses a CRISPR spacer (in blue) matching the protospacer encompassing the stop codon of *qatA*. Sequences of the G4S linker and the 6xHis tag are presented and denoted in magenta. The protospacer in *qatA* is underlined and shown in blue, while PAM motifs are in red. (B) Colony PCR analysis of the pHT-*qatA* transformants using primers *qat*_chkF and the specific His-R as depicted in panel A. The anticipated target bands are indicated by a magenta arrow. -, PCR amplification of a colony from the pAE1 transformant; M, DNA size marker. (C) Representative chromatograph of Sanger sequencing result verifying His-tagging in *qatA*. Sequenced DNA fragments were amplified from the pHT-*qatA* transformants using the *qat*_chkF/*qat*_chkR primer set as shown in panel A. *, stop codon. (D) Western blot analysis of His-tagged QatA using an antibody specifically against the His-tag peptide. Proteins of the predicted size (~72 kDa) were detected (lane labeled cQatA-HT). J481 and pQatA-HT, crude protein samples prepared from the wild-type J481 strain without or with the pQatA-HT plasmid, serving as negative and positive controls, respectively; M, protein size marker.

Using the primer set *qat*_chkF/His-R (Table S2), a PCR product of the expected size (518 bp) was consistently amplified from all 16 randomly selected transformants generated after electroporation of the pHT-*qatA* plasmid into Δ*mrr* cells. No such product was detected in the pAE1 transformant of Δ*mrr* (Fig. 4B). For verification, a DNA fragment spanning the edited locus was amplified with *qat*_chkF/*qat*_chkR primer pair (Table S2) and subsequently analyzed by Sanger sequencing. The results confirmed that all tested

colonies contained chromosomally His-tagged QatA (cQatA-HT; Fig. 4C), indicating a notably high editing efficiency of 100%. Moreover, the His-tagged QatA protein was detectable via western blot analysis. As shown in Fig. 4D, a protein band of approximately 72 kDa was observed in crude protein extracts derived from cQatA-HT cells and from J481 cells expressing the QatA-His fusion from a plasmid (pQatA-HT). This band was absent in the wild-type J481 strain. Collectively, the native type I-F CRISPR-mediated *in situ* protein tagging method offers a robust strategy for the *in vivo* functional characterization of target genes in *A. faecalis*.

## Establishment of a PheSv/4-CP counterselection system for efficient plasmid curing and hence rapid iterative CRISPR-based genome editing

Despite the high inherent efficacy of the CRISPR-based tool, the overall efficiency of iterative genome editing was significantly limited by the challenges associated with the efficient curing of the stable <u>v</u>ector for the construction of <u>g</u>enome <u>e</u>diting plasmids (pVGE) scaffold-based genome editing plasmids. To address this limitation, we aimed to develop a counterselection system capable of facilitating rapid plasmid curing, thereby expediting multi-round genome editing in *A. faecalis*. Counterselection based on PheS variants has emerged as one of the most convenient and widely adopted strategies due to its host-independent functionality. PheS is primarily responsible for phenylalanine aminoacylation. However, as initially illustrated in *E. coli*, a PheS* variant (containing the T251A and A294G mutations) preferentially aminoacylates a phenylalanine analog, 4-chloro-phenylalanine (4-CP), leading to dysfunctional proteins and ultimately cell death (33). Given the high conservation of *pheS* genes across bacterial species, counterselection based on PheS variants has been successfully implemented in a broad range of microorganisms (34–42). Leveraging these advantages, we sought to develop a PheS variant-based counterselection system in *A. faecalis*.

Amino acid sequence alignment revealed that the T259 and A306 residues in *Af*PheS (*A. faecalis* PheS; *D6I95_02200*) correspond to the T251 and A294 residues in *Ec*PheS (*E. coli* PheS) (33), respectively (Fig. 5A). Accordingly, substitutions T259A and A306G were introduced into *Af*PheS, generating a PheS variant, designated PheSv. Subsequently, a

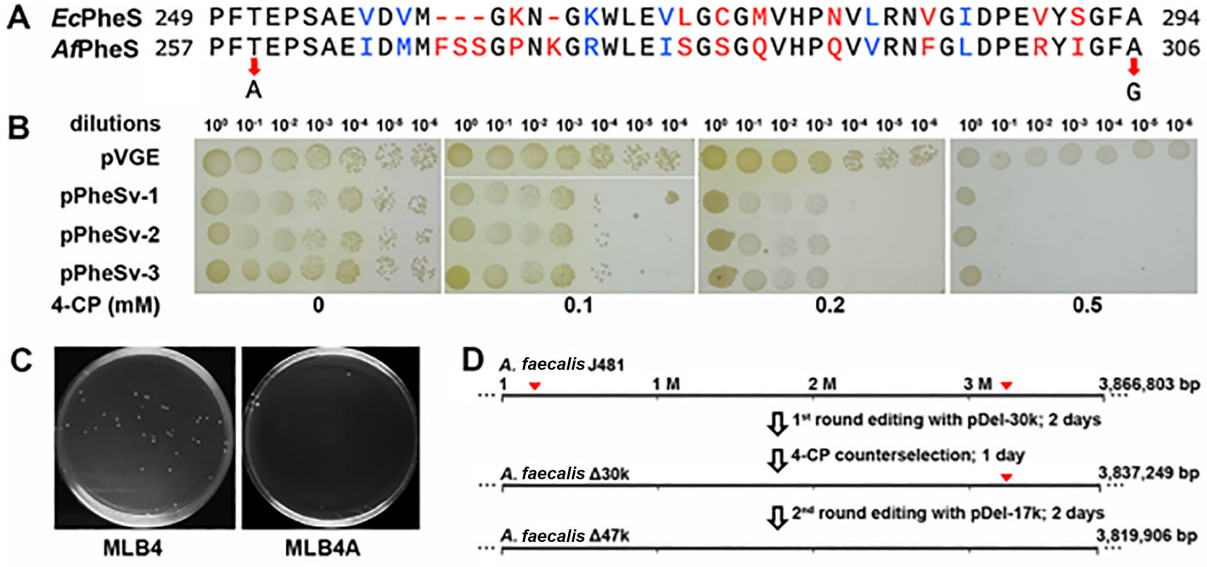

**FIG 5** Development of a counterselectable system for iterative genome editing. (A) Amino acid sequence alignment of PheS from *E. coli* and that from *A. faecalis*. The residuals being subjected to mutagenesis are indicated with red arrows. (B) 4-CP sensitivity of *A. faecalis* transformants. Cell cultures of transformants either carrying the cloning vector pVGE or the PheSv-expressing plasmid pPheSv were serially 10-fold diluted. Dilutions were spotted on agar plates containing 4-CP at the indicated concentrations. (C) Examination of the counterselection effect of 4-CP on curing the PheSv-bearing genome editing plasmid. (D) Schematic showcasing the iterative genome editing process. Editing loci were indicated with red arrowheads.

PheSv expression cassette was cloned to the pVGE vector, giving the pPheSv plasmid. *A. faecalis* cells carrying pPheSv, or pVGE as a reference, were grown on modified Luria-Bertani agar (MLBA) plates supplemented with 0, 0.1, 0.2, or 0.5 mM of 4-CP. The results showed that, in the absence of 4-CP, both pPheSv and pVGE transformants exhibited comparable growth across various dilutions. However, with increasing concentrations of 4-CP, growth retardation became apparent for the pPheSv transformants. When spotted on MLBA plates containing 0.5 mM of 4-CP (MLBA4), very few cells from the diluted pPheSv transformants were able to grow, whereas pVGE transformants continued to grow normally under identical conditions (Fig. 5B). Conceivably, bacterial survival during 4-CP exposure required pPheSv plasmid elimination, resulting in concomitant ampicillin sensitivity. These findings indicated that PheSv can serve as an effective counter-selectable marker in *A. faecalis*.

Subsequently, we employed 0.5 mM of 4-CP for counterselection to assist iterative genome editing. A PheSv expression cassette was inserted into the pDel-30k plasmid, yielding pDel-30kv. Electroporating pDel-30kv into the *A. faecalis* Δ*mrr* cells resulted in dozens of transformants, among which Δ30k mutants were confirmed. For a second round of genome editing, one of the Δ30k strains was inoculated in an MLB4 liquid medium and incubated for 24 h to facilitate plasmid curing. A $10^6$-fold dilution of the cells was spread onto either MLB4 or MLB4A agar plates. While colonies appeared on the MLB4 plate, cells failed to grow on the ampicillin-containing MLB4A plate (Fig. 5C), indicating successful yet rapid curing of the pDel-30kv plasmid.

Competent cells derived from the 4-CP-treated Δ30k mutant were then prepared for a second round of transformation using the pDel-17k plasmid for deleting a chromosomal stretch spanning genes *D6I95_14415* through *D6I95_14470*, which is part of a second prophage identified by PhiSpy (27). This allowed us to identify three strains from eight randomly selected colonies harboring the designed ~17 kb deletion, showing an editing efficiency of 37.5%. Therefore, the eventually generated Δ47k mutants each possess deletions of two chromosomal regions encompassing a total of 46 genes, that is, a ~30 kb fragment containing 34 genes and a ~17 kb fragment comprising 12 genes. It was noteworthy that two rounds of genome editing were completed within merely 5 days (Fig. 5D).

## DISCUSSION

In this study, we established the first functional native CRISPR-based genome editing platform in *A. faecalis*. This platform demonstrates substantially enhanced capabilities compared to existing type I-F CRISPR editing systems. While many systems achieve efficient single-gene knockouts, to the best of our knowledge, no previously reported type I-F CRISPR-Cas system in any species has accomplished precise deletion of genomic fragments approaching 30 kb nor achieved iterative deletions totaling 47 kb. The 47.6% efficiency for the 30 kb deletion and 37.5% for the subsequent 17 kb deletion significantly exceeds the performance of conventional two-step recombination in *A. faecalis*, which failed entirely for large fragments. Furthermore, the iterative editing speed, completing two complex large-scale deletions within 5 days via the integrated PheSv/4-CP counterselection system, is unprecedented. This efficiency fundamentally enhances the feasibility of complex multi-step genetic engineering in non-model bacteria.

Our advances hold substantial implications beyond fundamental genetic research, establishing *A. faecalis* as a promising synthetic biology chassis for environmental and industrial applications. Previous studies have shown that CRISPR-mediated chromosome targeting without homologous donor DNAs in *P. atrosepticum* resulted in uncontrolled genomic region loss (43). This phenomenon may be attributed to the presence of over 11 horizontally acquired islands in the *P. atrosepticum* chromosome (44), which contribute significantly to chromosome instability (45). In this work, the capability to rapidly delete prophages enables the rational design of "clean" chassis strains devoid of unnecessary or potentially detrimental DNA. This is crucial for enhancing genetic stability, reducing metabolic burden, and improving predictability in engineered pathways. The

integration of endogenous CRISPR targeting, R-M barrier removal, and rapid plasmid curing considerably streamlines the genetic engineering workflow in *A. faecalis*. This platform efficiency approaches the speed typically associated with highly tractable model organisms like *E. coli* and may surpass most other reported endogenous CRISPR-based editing systems in non-model microorganisms. This rapid iterative capability is paramount for advanced metabolic engineering and adaptive laboratory evolution studies.

While PCR and Sanger sequencing confirmed targeted deletion fidelity at the intended loci in all *A. faecalis* mutants, this study did not employ whole-genome sequencing (WGS) to systematically evaluate potential off-target mutations or genomic rearrangements induced by extended homologous recombination. Off-target effects represent a major challenge in CRISPR-Cas genome editing research. Methods have been developed for assessing off-target effects of the CRISPR-Cas9 system, which exhibit substantial bias toward eukaryotic organisms (46). Given that eukaryotes possess the non-homologous end joining (NHEJ) pathway, they can efficiently repair DNA breaks in an error-prone manner, resulting in indels at both on-target and off-target sites. These genetic alterations can be detected via WGS. However, most bacteria, including *A. faecalis*, lack the NHEJ system and rely primarily on homology-directed repair for accurate DNA break resolution. Consequently, in bacteria, CRISPR-mediated targeting of the host genome without homologous donor DNA typically leads to cell death. Although previous studies have reported CRISPR-mediated chromosome targeting without homologous donor DNAs resulting in genomic region loss in certain bacteria, this outcome was not observed in *A. faecalis*, where near-complete cell death occurred, potentially due to its relatively compact chromosome with essential genes distributed throughout. Thus, assessing off-target effects via WGS in *A. faecalis* presents considerable challenges. Nevertheless, future studies implementing large-scale deletions, particularly for synthetic biology applications requiring genomic stability, should incorporate WGS analysis of edited strains to exclude unintended mutations and confirm the absence of adaptive costs or replication abnormalities. Encouragingly, the robust growth observed in the Δ47k mutants under laboratory conditions (data not shown) provides preliminary evidence for viability post-deletion.

Conclusively, this work offers a significant reference for developing efficient genetic tools in numerous industrially or environmentally relevant prokaryotes where heterologous CRISPR-Cas systems exhibit toxicity or transformation is inefficient. The core principles demonstrated—integrating the harnessing of intrinsic mechanisms for precision genome targeting with the overcoming of host defense barriers to enhance efficiency—provide a robust framework for unlocking the genetic potential of myriad underutilized prokaryotes in biotechnology and environmental science.

## MATERIALS AND METHODS

### Strains, growth conditions, and electroporation transformation of *A. faecalis*

*A. faecalis* J481 and derivatives constructed in this work were listed in Table S1. Cells were grown at 30°C in a modified Luria-Bertani (MLB) medium (10 g/L tryptone, 5 g/L yeast extract, and 30 g/L NaCl). If required, ampicillin or kanamycin was supplemented to the final concentrations of 100 µg/mL and 50 µg/mL for *A. faecalis* and *Escherichia coli*, respectively. For competent cell preparation, *A. faecalis* cells were harvested at an $OD_{600}$ of 0.6, measured using the Nano-800+ ultra-microspectrophotometer (Shanghai JP Analytical Instrument Co., Ltd., Shanghai, China), and washed twice with 10% glycerol (vol/vol). Plasmid transformation was carried out via electroporation (0.1 cm gap cuvettes, 1.6 kV, 200 W, and 25 µF) using Bio-Rad Gene Pulser (Bio-Rad, Hercules, CA, USA). Electroporated cells were incubated in the MLB medium for 3 h at 30°C prior to plating.

## Construction of plasmids

An *A. faecalis*-*E. coli* shuttle vector, pAE1, was created by replacing the ColE1 origin of replication of pUC19 with the replicon of the broad-host pBBR1 plasmid (47). Interference plasmids were constructed with two spacers, Spacer1 of Array2 (A1S1) and Spacer6 of Array2 (C2S6). Oligonucleotides were designed to bear the entire sequences of the selected spacers following a 5′-CCC-3′ PAM or the last three nucleotides of the repeat (5′-AAA-3′). Each set of oligonucleotides was annealed by first being heated to 95°C for 5 min and subsequently cooled down gradually to room temperature, followed by double digestion using *Eco*RI and *Bam*HI. These DNA fragments were ligated with the *Eco*RI and *Bam*HI linearized vector, yielding the interference plasmids (pInt plasmids) and the corresponding reference plasmids (pRef plasmids).

Based on pAE1, a basal pVGE was constructed. First, a DNA fragment (synthetic DNA 1; syn1), consisting of the leader sequence of the chromosomal CRISPR Array2 as a promoter and two CRISPR repeats that were spaced by two oppositely oriented *Bsa*I recognition sequences, was synthesized from CWBIO Biotechnology Co., Ltd. (Taizhou, China) and inserted into pAE1 after *Xma*I and *Sac*I digestions, yielding the pVGE plasmid vector. Digestion of pVGE with *Bsa*I gave a linearized plasmid having protruding termini. Double-stranded spacers were produced to contain protruding ends complementary to those in the linearized pVGE by annealing oligonucleotides as follows: being heated to 99°C for 10 min, cooling down gradually to room temperature, and then ligating a spacer with the linearized pVGE, which generated an artificial CRISPR-bearing plasmid. To yield the all-in-one genome editing plasmids, homologous arms (repair templates) were amplified from the genome of *A. faecalis* J481 with specific primer sets (Table S2) or synthesized in the case of *in situ* His-tagging plasmid (syn2) and ligated to their cognate self-targeting plasmids using the ClonExpress Ultra One Step Cloning Kit V3 (Vazyme Biotech Co., Ltd., Nanjing, China) following the instructions. For counterselection, the *pheS* mutant (*pheSv*) was generated through splicing and overlap extension PCR (48) and inserted into either a genome editing plasmid or the pVGE vector at *Hind*III and *Sal*I sites.

Expression plasmids of Avs2 and QatA were constructed based on pSpy (GenBank acc. no. PP457283), while the *E. coli* lambda phage portal protein (*B* gene) was constructed based on pAE1. The *avs2* gene, along with its native promoter, was amplified from the genome of *A. faecalis* J481, while the *qatA* gene, along with its native promoter and a 3′-end G4S-6xHis coding sequence, was amplified from the genome of the cQatA-HT strain, which were individually cloned to the pSpy vector at *Sac*I and *Xba*I sites, generating pAvs2 and pQatA-HT, respectively. An expression cassette of the lambda phage *B* gene (syn3) was synthesized and cloned to pAE1 after *Sal*I and *Eco*RI digestions, giving pELB.

All plasmids were listed in Table S1. All oligonucleotides, including syn1, syn2, and syn3, were synthesized from CWBIO Biotechnology Co., Ltd. (Taizhou, China) and listed in Table S2. Complete sequences of shuttle vectors were included in Table S3. Restriction enzymes were purchased from Vazyme Biotech Co., Ltd. (Nanjing, China), and T5 exonuclease from New England Biolabs (Beijing) Ltd. (Beijing, China).

## Construction and screening of mutant strains

Genome editing plasmids for chromosomal modifications were individually introduced into *A. faecalis* cells. Electroporated cells were spread on MLBA plates containing ampicillin at a final concentration of 100 µg/mL and incubated at 30°C until colonies were seen. Mutant candidates were screened by colony PCR using primers listed in Table S2. PCR products were subjected to agarose gel electrophoresis, and the separated DNA fragments were intercalated with ethidium bromide for visualization using the QuickGel 6200 Imaging Systems (Monad Biotech Co., Ltd., Suzhou, China). PCR products with predicted sizes were sent for Sanger sequencing confirmation.

## 4-CP sensitivity assay and curing of genome editing plasmids

To assay the sensitivity of *A. faecalis* strains to 4-CP, a growth inhibition test was done. Overnight cultures of *A. faecalis* strains were diluted in fresh MLB medium and grown to an $OD_{600}$ of 0.4. Then, each of the cultures was serially 10-fold diluted up to $10^{-6}$, and 5 µL of each dilution was spotted onto MLBA plates where different concentrations of 4-CP were supplemented. The growth of each strain was photographically recorded after 72 h incubation at 30°C.

For genome-editing plasmid curing, cells of the transformants were incubated in an MLBA medium and then spread onto an agar plate containing 0.5 mM 4-CP with (MLB4A) or without ampicillin (MLB4). Cells that have formed colonies on the MLB4 plate are regarded as those that lost the genome editing plasmid.

## Western blotting

The protocol used for western blotting was modified slightly based on previous studies (18, 28). *A. faecalis* J481 strains with or without pQatA-HT, as well as the cQatA-HT strain, were cultured in MLB medium. Cells were harvested by centrifugation when $OD_{600}$ reached 0.6, followed by suspension in 50 mM phosphate buffer for sonication lysis. The crude protein extracts were fractioned on 12% SDS-PAGE and transferred onto a nylon membrane using the Semi-Dry Electrophoretic Transfer Cell System (Bio-Rad, Hercules, CA, USA). The membrane was incubated with a hybridization buffer containing a mouse anti-His antibody. The antibody bound to the His-tagged proteins was then recognized by a secondary goat anti-mouse IgG antibody. The results were then visualized by chemiluminescent detection using the Clarity Western ECL substrate (Bio-Rad, Hercules, CA, USA) and recorded with the Amersham Imager 600 device (GE Healthcare, Seoul, South Korea).

## ACKNOWLEDGMENTS

We acknowledge the financial support provided by Wuhan mjBio Biotechnology Co., Ltd, Wuhan, China. W.P. and Y. Zheng also express their sincere appreciation for the support received from the State Key Laboratory of Biocatalysis and Enzyme Engineering.

## AUTHOR AFFILIATIONS

[1]State Key Laboratory of Biocatalysis and Enzyme Engineering, Hubei Engineering Research Center for Microbial Cell Factories, Hubei Key Laboratory of Industrial Microbiology, School of Life Sciences, Hubei University, Wuhan, People's Republic of China
[2]College of Life Science and Technology, Wuhan Polytechnic University, Wuhan, People's Republic of China

## AUTHOR ORCIDs

Yanli Zheng ⓘ http://orcid.org/0000-0001-5923-773X
Wenfang Peng ⓘ http://orcid.org/0000-0001-9073-1509

## AUTHOR CONTRIBUTIONS

Wanting Cheng, Conceptualization, Data curation, Investigation, Writing – original draft | Jiaxin Li, Conceptualization, Data curation, Investigation, Writing – original draft | Lei Lei, Validation, Writing – review and editing | Yuxuan Zhu, Validation, Writing – review and editing | Siqi Luo, Validation, Writing – review and editing | Xueqing Wang, Formal analysis, Writing – review and editing | Qinghui Zhang, Data curation, Investigation, Validation | Miaomiao Cao, Data curation, Investigation, Validation | Yanli Zheng, Conceptualization, Supervision, Validation, Writing – review and editing | Wenfang Peng,

Conceptualization, Funding acquisition, Supervision, Writing – original draft, Writing – review and editing

## DATA AVAILABILITY

The authors declare that the main data supporting the findings of this work are available within the article and its supplemental material files or from the corresponding authors upon reasonable request.

## ADDITIONAL FILES

The following material is available online.

### Supplemental Material

**Supplemental material (Spectrum02786-25-s0001.docx).** Fig. S1 and S2; Tables S1 to S3.

### Open Peer Review

**PEER REVIEW HISTORY (review-history.pdf).** An accounting of the reviewer comments and feedback.

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
