## [Reviewer comments · Microbiology Spectrum]

Microbiology Spectrum

Unlocking genome engineering in *Alcaligenes faecalis* by exploiting its native Type I-F CRISPR-Cas

Wanting Cheng, Jiaxin Li, Lei Lei, Yuxuan Zhu, Siqi Luo, Xueqing Wang, Qinghui Zhang, Yanli Zheng, and Wenfang Peng

Corresponding Author(s): Wenfang Peng, Hubei University

Review Timeline:

Submission Date:	September 4, 2025
Editorial Decision:	October 30, 2025
Revision Received:	December 3, 2025
Editorial Decision:	December 16, 2025
Revision Received:	December 23, 2025
Accepted:	January 23, 2026

Editor: Jeffrey Gardner

Reviewer(s): Disclosure of reviewer identity is with reference to reviewer comments included in decision letter(s). The following individuals involved in review of your submission have agreed to reveal their identity: Zhi-Qiang Xiong (Reviewer #2)

Transaction Report:

DOI: <https://doi.org/10.1128/spectrum.02786-25>

Re: Spectrum02786-25 (**Unlocking genome engineering in *Alcaligenes faecalis* by exploiting its native Type I-F CRISPR-Cas**)

Dear Dr. Wenfang Peng:

Thank you for the privilege of reviewing your work. Below you will find my comments, instructions from the Spectrum editorial office, and the reviewer comments.

Revision Guidelines

Sincerely,
Jeffrey Gardner
Editor
Microbiology Spectrum

Reviewer #1 (Comments for the Author):

The authors consider the currently available genetic manipulation methods (two-steps editing) for *Alcaligenes faecalis* are time-consuming for multiple-gene knockouts and are nearly impractical for large-fragment deletions. Therefore, the authors aimed to develop a CRISPR-Cas-based strategy. They first attempted to use the commonly employed CRISPR-Cas9 system, but failed due to its toxicity. Consequently, they explored the use of the endogenous type I-F CRISPR-Cas system in *A. faecalis*. They

successfully deployed this system, achieving two successive deletions of 30 kb and 17 kb within five days. With the aid of the counter-selectable marker pheS*, the editing plasmid could be efficiently cured after each editing round, facilitating multiple cycles of genome editing.

Major comments

1. The authors' main claim is that this CRISPR-based method achieves longer deletions in less time compared to two-steps editing approaches. To convincingly support this point, a direct comparison should be made under identical condition at the same locus and with the same deletion size.
2. The authors report that deleting mrr improved transformation efficiency. How did the authors delete mrr? After deletion, the efficiency is described as yielding "hundreds of transformants," whereas before deletion it was 242 transformants. What is the precise fold increase? In addition, why was only the type IV R-M system targeted? Does this strain exclusively possess type IV R-M?
3. No knock-in (gene integration) experiments were demonstrated.
4. Both knock-out and knock-in strategies should be validated with at least three representative cases to demonstrate robustness.

Minor comments

1. It is unclear whether this CRISPR-Cas system was identified by the authors themselves or had been previously characterized and only deployed here.
2. In Figure 1, the design rationale of pInt (and pRef) is not sufficiently explained.
3. Line 139: the word "hijack" does not appear appropriate in this context.
4. Line 140: The statement "We envisioned that self-targeting crRNAs generated from the artificial CRISPR would direct the CRISPR-Cas effector complex to the target sequence" is difficult to follow. The meaning of "self-targeting crRNAs" should be clarified.
5. Please explain the rationale for selecting 500 bp homology arms.
6. Line 157: the word "via" should be italicized.

Reviewer #2 (Comments for the Author):

Chen et al. successfully developed and optimized a genome editing toolkit based on the endogenous Type I-F CRISPR-Cas system of *Alcaligenes faecalis*, achieving efficient single-gene knockout and precise iterative deletion of large genomic fragments in this strain for the first time. It provides key technical support for the field of environmental microbial genetic engineering and exhibits significant innovation and application value. However, the article still has room for improvement in terms of content depth, experimental completeness, and expression standardization. Specific revision suggestions are as follows:

1. Although the study emphasizes "the first establishment of an endogenous CRISPR genome editing platform in *Alcaligenes faecalis*", the main text does not fully elaborate on the differentiated innovation of this research compared with existing studies on Type I-F CRISPR-Cas systems. It is recommended to add horizontal comparative analysis in the Introduction and Discussion sections to clarify the breakthrough progress of this study in terms of editing efficiency, deletable fragment length, and iterative editing speed.
2. For the 30 kb and 47 kb large-fragment deletion experiments, no whole-genome off-target effect analysis was performed, nor was it reported whether continuous editing caused genomic rearrangement, replication abnormalities, or adaptive costs. It is recommended to supplement off-target risk assessment experiments to verify that there are no additional mutations in the bacterial genome except for the target region.
3. The discussion on the systematic significance of large-fragment deletion and rapid plasmid curing technology, as well as their application prospects in synthetic biology, is relatively brief. It is recommended to expand the discussion content and conduct multi-dimensional comparisons with other reported endogenous CRISPR systems to strengthen the interpretation of research value.
4. Non-Standard Expression and Format: (1) The expression "hundreds-fold lower" in Figure 1D lacks specific numerical support. It is recommended to supplement accurate fold-change data in the figure legend or main text to improve the credibility of the results; (2) The spelling error "faecalis" appears multiple times throughout the article, and the correct spelling should be "faecalis"; (3) Inconsistent use of terms, such as "PheSv" sometimes expressed as "PheS variant", which needs to be unified; (4) There are duplicate paragraphs in the "Materials and Methods" section, and redundant content should be deleted to ensure a clear structure; (5) The supplementary table information is not fully cited. The strains, primers and other information listed in Tables S1 and S2 should be clearly marked and cited in the methodology section to enhance experimental reproducibility.

Reviewer #1 (Comments for the Author):

*The authors consider the currently available genetic manipulation methods (two-steps editing) for *Alcaligenes faecalis* are time-consuming for multiple-gene knockouts and are nearly impractical for large-fragment deletions. Therefore, the authors aimed to develop a CRISPR-Cas-based strategy. They first attempted to use the commonly employed CRISPR-Cas9 system, but failed due to its toxicity. Consequently, they explored the use of the endogenous type I-F CRISPR-Cas system in *A. faecalis*. They successfully deployed this system, achieving two successive deletions of 30 kb and 17 kb within five days. With the aid of the counter-selectable marker *pheS**, the editing plasmid could be efficiently cured after each editing round, facilitating multiple cycles of genome editing.*

Major comments

1. The authors' main claim is that this CRISPR-based method achieves longer deletions in less time compared to two-steps editing approaches. To convincingly support this point, a direct comparison should be made under identical condition at the same locus and with the same deletion size.

We fully acknowledge that an ideal direct comparison at the same locus targeting the same deletion length would be highly valuable. However, a critical limitation prevented this exact experiment: using the conventional two-step editing approach on the same long deletion targets consistently failed to yield any viable transformants after screening. This outcome highlights a fundamental limitation of the two-step method for large-scale deletions in this bacterium. While direct side-by-side comparison under identical conditions wasn't possible due to failed transformation in the conventional approach, the consistent success of our CRISPR-based method where the two-step approach failed supports our claim of superior efficacy specifically for deletions of large genomic stretches.

*2. The authors report that deleting *mrr* improved transformation efficiency. How did the authors delete *mrr*? After deletion, the efficiency is described as yielding "hundreds of transformants," whereas before deletion it was 242 transformants. What is the precise fold increase? In addition, why was only the type IV R-M system targeted? Does this strain exclusively possess type IV R-M?*

We sincerely appreciate your questions. The *mrr* gene was successfully deleted using the native Type I-F CRISPR-Cas, albeit with suboptimal editing efficiency yet achieving the desired outcome. By utilizing the *mrr* deletion mutant as a host strain, we obtained "hundreds of transformants", which stands in stark contrast to the merely 1-2 transformants typically yielded in the wild-type strain. We have modified the description as "Remarkably, hundreds of transformants were obtained, which is in sharp contrast to the yield of merely 1-2 transformants when hosted by the wild-type J481 cells..."

Throughout the manuscript, transformation efficiency was quantified as colony-forming units per microgram of plasmid DNA (CFU/ μ g), rather than the absolute number of transformants. While the quantities of transformants remained comparable, significant variations in transformation efficiency were

observed. For instance, the introduction of approximately 0.5 µg of plasmid DNA yielded 242 transformants, establishing an average transformation efficiency of 491 ± 52 CFU/µg plasmid DNA. We had to use >0.5 µg DNAs of genome editing plasmids as lower DNA amount gave no or very few transformants. To enhance clarity, the relevant text has been revised as follows:

Following introducing the pKO-*avs2* plasmid into the wild-type J481 cells *via* electroporation, a total of 242 transformants formed colonies on an ampicillin selection plate, **corresponding to an average transformation efficiency of 491 ± 52 CFU/µg plasmid DNA.**" Comparative analysis of transformation efficiencies between the mutant and wild-type strains was conducted using the pAE1 shuttle vector, with both values expressed in CFU/µg plasmid DNA.

We actually identified three R-M systems in the bacterium and demonstrated the dominant function of the Type IV system in preventing foreign DNAs, which we have included in the main text as follows:

Three R-M systems were identified in *A. faecalis* J481 using DefenseFinder, including (i) a Type I module encoding *hsdR1* (D6I95_14420), *hsdS* (D6I95_14425), *hsdR2* (D6I95_14430), and *hsdM* (D6I95_14435); (ii) a Type III system comprising a restriction endonuclease (REase, D6I95_06790) and a methyltransferase (MTase, D6I95_06785); and (iii) a Type IV *mrr* element (D6I95_00770). Transformation experiments were conducted using 10 ng of the *A. faecalis*-*E. coli* shuttle plasmid pAE1, which was independently prepared from *E. coli* DH5α (*dam*⁺*dcm*⁺) and Trans110 (*dam*⁻*dcm*⁻) strains. The transformation efficiency achieved with pAE1 prepared from DH5α cells was $(2.22 \pm 0.31) \times 10^5$ CFU/µg plasmid DNA, demonstrating a 76-fold reduction compared to that obtained with pAE1 extracted from Trans110 cells. This significant difference indicates that the IV R-M system, which has been shown to specifically target modified DNA in other bacterial species, plays a crucial role in foreign DNA restriction.

3. *No knock-in (gene integration) experiments were demonstrated.*

4. *Both knock-out and knock-in strategies should be validated with at least three representative cases to demonstrate robustness.*

Thank you very much for your recommendations (3 & 4). In a parallel study, we are performing mechanistic analyses of antiviral defense systems including QatABCD, Avs2 and 3PH in *A. faecalis*, for which we have constructed knockout strains of these systems and performed *in situ* tagging of the *qatA* gene (knock-in) using the optimized native CRISPR-Cas genome editing system established in this work. All the editing options were accomplished with considerably high editing efficiencies, specifically, 96% for *qatABCD* operon deletion, 100% for *vap1* deletion, 93.25% for 3PH deletion, and 100% for *in situ* His-tagging of *qatA*. These collective results robustly demonstrate the reliability of the developed Type I-F CRISPR-based genome editing system. However, to avoid redundant presentation of analogous gene knockout outcomes, we have included only the result corresponding to the *in situ* His-tagging of *qatA* (in main text, Section "Implementation of the Type I-F CRISPR-based tool for *in situ* gene tagging" and Fig. 4 in the revised manuscript).

Implementation of the Type I-F CRISPR-based tool for *in situ* gene tagging. We subsequently evaluated the efficacy of this Type I-F CRISPR-based tool for performing *in situ* gene tagging in *A. faecalis*, thereby facilitating *in vivo* functional investigation of target genes. To this end, we designed and constructed the genome editing plasmid, pHT-*qatA*, for *in situ* His-tagging of the *qatA* gene (*D6I95_14445*), which encodes the NTPase component of the QatABCD phage defense system.

A critical challenge in this editing lies in the necessity to modify the sequence after tag insertion to render the protospacer unrecognizable by the CRISPR-Cas system, thus ensuring cell viability. Through careful examination of the sequences surrounding the *qatA* stop codon, we identified a 5'-CCC-3' PAM located on the non-coding strand. Consequently, the 32-nt sequence immediately downstream of the PAM was considered as a protospacer. A tag coding sequence (G4S-6xHis) was strategically designed to be inserted immediately upstream of the *qatA* stop codon, thereby interrupting the protospacer (**Fig. 4A**).

Using the primer set *qat_chkF/His-R* (**Table S2**), a PCR product of the expected size (518 bp) was consistently amplified from all 16 randomly selected transformants generated after electroporation of the pHT-*qatA* plasmid into Δmrr cells. No such product was detected in the pAE1 transformant of Δmrr (**Fig. 4B**). For verification, a DNA fragment spanning the edited locus was amplified with *qat_chkF/qat_chkR* primer pair (**Table S2**) and subsequently analyzed by Sanger sequencing. The results confirmed that all tested colonies contained chromosomally His-tagged QatA (cQatA-HT) (**Fig. 4C**), indicating a notably high editing efficiency of 100%. Moreover, the His-tagged QatA protein was detectable *via* Western blot analysis. As shown in **Fig. 4D**, a protein band of approximately 72 KDa was observed in crude protein extracts derived from cQatA-HT cells and from J481 cells expressing the QatA-His fusion from a plasmid (pQatA-HT). This band was absent in the wild-type J481 strain. Collectively, the native Type I-F CRISPR-mediated *in situ* protein tagging method offers a robust strategy for the *in vivo* functional characterization of target genes in *A. faecalis*.

Fig. 4 Implementation of Type I-F CRISPR-based tool for *in situ* protein tagging. (A) Schematic diagram illustrating the strategy for *in situ* His-tagging of QatA. The genome editing plasmid pHT-*qatA* expresses a CRISPR spacer (in blue) matching the protospacer encompassing the stop codon of *qatA*. Sequences of the G4S linker and the 6xHis tag are presented and denoted in magenta. The protospacer in *qatA* is underlined and shown in blue, while PAM motifs are in red. (B) Colony PCR analysis of the pHT-*qatA* transformants suing primers *qat_chkF* and the specific His-R as depicted in (A). The anticipated target bands are indicated by a magenta arrow. -, PCR amplification of a colony from the pAE1 transformant; M, DNA size marker. (C) Representative chromatograph of Sanger sequencing result verifying His-tagging in *qatA*. Sequenced DNA fragments were amplified from the pHT-*qatA* transformants using the *qat_chkF*/*qat_chkR* primer set as shown in (A). *, stop codon. (D) Western blot analysis of His-tagged QatA using an antibody specifically against the His-tag peptide. Proteins of the predicted size (~72 kDa) was detected (lane labeled cQatA-HT). J481 and pQatA-HT, crude protein samples prepared from the wild-type J481 strain without or with the pQatA-HT plasmid, serving as negative and positive controls, respectively; M, protein size marker.

Minor comments

1. *It is unclear whether this CRISPR-Cas system was identified by the authors themselves or had been previously characterized and only deployed here.*

We have added description in the INTRODUCE section stating “... **However, systematic investigations exploring native CRISPR-Cas systems for genome editing in *A. faecalis* are still notably absent. In this study, we functionally characterized the native Type I-F CRISPR-Cas of *A. faecalis* J481 and engineered it into a versatile genome editing platform...**”

2. *In Figure 1, the design rationale of pInt (and pRef) is not sufficiently explained.*

A panel has been added to Figure 1D explaining the design rationale of pInt and pRef.

3. *Line 130: the word "hijack" does not appear appropriate in this context.*

Now the description is present as “... under **standard laboratory** conditions, **thereby presenting a viable candidate for genome editing applications.**”

4. *Line 140: The statement "We envisioned that self-targeting crRNAs generated from the artificial CRISPR would direct the CRISPR-Cas effector complex to the target sequence" is difficult to follow. The meaning of "self-targeting crRNAs" should be clarified.*

Revised as “We envisioned that crRNAs generated from the artificial CRISPR would direct the CRISPR-Cas effector complex to the **protospacer sequence in *avs2***, exerting cut of the chromosome.”

5. *Please explain the rationale for selecting 500 bp homology arms.*

We provided additional experimental data in **Fig. S1** explaining the rationale for selecting 500 bp homology arms.

6. *Line 157: the word "via" should be italicized.*

Italicized.

Reviewer #2 (Comments for the Author):

*Cheng et al. successfully developed and optimized a genome editing toolkit based on the endogenous Type I-F CRISPR-Cas system of *Alcaligenes faecalis*, achieving efficient single-gene knockout and precise iterative deletion of large genomic fragments in this strain for the first time. It provides key technical support for the field of environmental microbial genetic engineering and exhibits significant innovation and application value. However, the article still has room for improvement in terms of content depth, experimental completeness, and expression standardization. Specific revision suggestions are as follows:*

1. *Although the study emphasizes "the first establishment of an endogenous CRISPR genome editing platform in *Alcaligenes faecalis*", the main text does not fully elaborate on the differentiated innovation of this research compared with existing studies on Type I-F CRISPR-Cas systems. It is recommended to add horizontal comparative analysis in the Introduction and Discussion sections to clarify the*

breakthrough progress of this study in terms of editing efficiency, deletable fragment length, and iterative editing speed.

Thank you very much for your suggestive recommendation. We have added content as recommended as follows:

In INTRODUCTION

Specifically, Type I-F CRISPR-Cas systems have been efficiently employed for targeted genome editing in *Pseudomonas aeruginosa*, *Zymomonas mobilis*, and *Acinetobacter baumannii*, and more recently for genome engineering and gene repression in *P. chlororaphis*. Furthermore, the Type I-F CRISPR-Cas of *Pectobacterium atrosepticum* has been implicated in facilitating the remodeling or deletion of pathogenicity islands when targeting non-essential chromosomal regions. These advancements have unveiled novel prospects for repurposing *A. faecalis*-derived CRISPR-Cas as intrinsic genome editing tools. However, systematic investigations exploring endogenous CRISPR-Cas systems for genome editing in *A. faecalis* are still notably absent.

In DISCUSSION

This platform demonstrates substantially enhanced capabilities compared to existing Type I-F CRISPR editing systems. While many systems achieve efficient single-gene knockouts, to the best of our knowledge, no previously reported Type I-F CRISPR-Cas system in any species has accomplished precise deletion of genomic fragments approaching 30-kb, nor achieved iterative deletions totaling 47-kb...

2. For the 30 kb and 47 kb large-fragment deletion experiments, no whole-genome off-target effect analysis was performed, nor was it reported whether continuous editing caused genomic rearrangement, replication abnormalities, or adaptive costs. It is recommended to supplement off-target risk assessment experiments to verify that there are no additional mutations in the bacterial genome except for the target region.

As recommended, we have carefully discussed this in the “DISCUSS” section as follows: While PCR and Sanger sequencing confirmed targeted deletion fidelity at the intended loci in all *A. faecalis* mutants; this study did not employ whole-genome sequencing (WGS) to systematically evaluate potential off-target mutations or genomic rearrangements induced by extended homologous recombination. Off-target effects represent a major challenge in CRISPR-Cas genome editing research. Methods have been developed for assessing off-target effects of the CRISPR-Cas9 system; which, they exhibit substantial bias toward eukaryotic organisms. Given that eukaryotes possess the non-homologous end joining (NHEJ) pathway, they can efficiently repair DNA breaks in an error-prone manner, resulting in indels at both on-target and off-target sites. These genetic alterations can be detected *via* WGS. However, most bacteria, including *A. faecalis*, lack the NHEJ system and rely primarily on homology-directed repair for accurate DNA break resolution. Consequently, in bacteria, CRISPR-mediated targeting of the host genome without homologous

donor DNA typically leads to cell death. Although previous studies have reported CRISPR-mediated chromosome targeting without homologous donor DNAs resulting in genomic region loss in certain bacteria, this outcome was not observed in *A. faecalis*, where near-complete cell death occurred, potentially due to its relatively compact chromosome with essential genes distributed throughout. Thus, assessing off-target effects *via* WGS in *A. faecalis* presents considerable challenges. Nevertheless, future studies implementing large-scale deletions, particularly for synthetic biology applications requiring genomic stability, should incorporate WGS analysis of edited strains to exclude unintended mutations and confirm the absence of adaptive costs or replication abnormalities. Encouragingly, the robust growth observed in the $\Delta 47k$ mutants under laboratory conditions (data not shown) provides preliminary evidence for viability post-deletion.

3. The discussion on the systematic significance of large-fragment deletion and rapid plasmid curing technology, as well as their application prospects in synthetic biology, is relatively brief. It is recommended to expand the discussion content and conduct multi-dimensional comparisons with other reported endogenous CRISPR systems to strengthen the interpretation of research value.

Thank you very much for your valuable comments. Have discussed it in the "DISCUSS" section as: **In this study, we established the first functional native CRISPR-based genome editing platform in *A. faecalis*. This platform demonstrates substantially enhanced capabilities compared to existing Type I-F CRISPR editing systems. While many systems achieve efficient single-gene knockouts, to the best of our knowledge, no previously reported Type I-F CRISPR-Cas system in any species has accomplished precise deletion of genomic fragments approaching 30-kb, nor achieved iterative deletions totaling 47-kb. The 47.6% efficiency for the 30-kb deletion and 37.5% for the subsequent 17-kb deletion significantly exceed the performance of conventional two-step recombination in *A. faecalis*, which failed entirely for large fragments. Furthermore, the iterative editing speed, completing two complex large-scale deletions within 5 days *via* the integrated PheSv/4-CP counterselection system, is unprecedented. This efficiency fundamentally enhances the feasibility of complex multi-step genetic engineering in non-model bacteria.**

4. Non-Standard Expression and Format:(1) The expression "hundreds-fold lower" in Figure 1D lacks specific numerical support. It is recommended to supplement accurate fold-change data in the figure legend or main text to improve the credibility of the results;(2) The spelling error "feacelis" appears multiple times throughout the article, and the correct spelling should be "faecalis";(3) Inconsistent use of terms, such as "PheSv" sometimes expressed as "PheS variant", which needs to be unified;(4) There are duplicate paragraphs in the "Materials and Methods" section, and redundant content should be deleted to ensure a clear structure;(5) The supplementary table information is not fully cited. The strains, primers and other information listed in Tables S1 and S2 should be clearly marked and cited in the methodology section to enhance experimental reproducibility.

Thank you very much for your valuable recommendations, in the revised version, (1) fold-change was indicated in the main text, which is present as "...it was observed that the transformation rates associated with pInt-A1S1 or pInt-A2S6 were >3,700-fold lower than those with the reference plasmids (**Figure 1D**) ..."

(2) all wrong spelling ("*faecelis*") have been corrected as "*faecelis*". (3) the constructed PheS variant is designated as PheSv, making them uniform. (4) the redundant content in the "Materials and Methods" section has been removed. and (5) revised as suggested.

Re: Spectrum02786-25R1 (**Unlocking genome engineering in *Alcaligenes faecalis* by exploiting its native Type I-F CRISPR-Cas**)

Dear Dr. Wenfang Peng:

Thank you for the privilege of reviewing your work. Below you will find my comments, instructions from the Spectrum editorial office, and the reviewer comments.

While one Reviewer did not have additional comments, another was still unsatisfied with the work in the revision. As the editor, I reviewed your resubmission and concur that you need to acknowledge the lack of replication and that Reviewer comments cannot be ignored. There are also other concerns and inconsistencies that were brought up by the Reviewer that need to be addressed. In the R2 resubmission (if you choose to resubmit), all these issues must be addressed, or the R2 version of the manuscript is at risk for rejection.

Revision Guidelines

Sincerely,
Jeffrey Gardner
Editor
Microbiology Spectrum

Reviewer #1 (Comments for the Author):

Although the authors have responded to each of my questions, several key major concerns remain unresolved, as detailed below.

1. The authors state that: "using the conventional two-step editing approach on the same long deletion targets consistently failed to yield any viable transformants after screening". Therefore, they could not directly compare the two-step approach with the method developed in this manuscript. The reviewer finds this explanation insufficient.

Even if large-fragment editing is challenging for the two-step method, the authors could still perform comparative experiments using shorter fragment lengths to validate one of the claimed advantages of the proposed method, namely reduced time requirements. In addition, the authors should clearly specify from what fragment size onward traditional single- or double-crossover approaches fail to generate clones, thereby defining the practical boundary of applicability for these methods.

At present, the manuscript does not provide a clear or quantitative description addressing these points.

2. In the response, the authors state that the relevant text was revised to:

"Remarkably, hundreds of transformants were obtained, which is in sharp contrast to the yield of merely 1-2 transformants when hosted by the wild-type J481 cells..."

However, in the revised manuscript, the actual wording is:

"Remarkably, more than 500 transformants were observed, which is in sharp contrast to the yield of merely 1-2 transformants when hosted by the wild-type J481 cells."

No line numbers were provided in the response, making it difficult for me to locate the corresponding revisions in the updated manuscript.

Moreover, the data indicate that 242 transformants were obtained using pKO-avs2 prior to mrr deletion, whereas 500 transformants were obtained after mrr deletion, corresponding to only an approximately twofold increase, which does not appear particularly substantial.

The authors argue that the 242 transformants were obtained using 0.5 µg of DNA. It remains unclear whether the comparison described in Lines 194-195 was conducted using equivalent amounts of DNA, and if so, what the exact DNA quantities were. Moreover, 0.5 µg DNA is not an unusually high amount in many bacterial CRISPR-Cas editing experiments.

3. The reviewer recommended that the authors demonstrate the general applicability of the tool by presenting both knockout and knock-in experiments, and by including at least three different target genes, preferably of different sizes.

In their response, the authors state that they achieved: "96% for qatABCD operon deletion, 100% for vap1 deletion, 93.25% for 3PH deletion, and 100% for in situ His-tagging of qatA".

However, the authors further state:

"to avoid redundant presentation of analogous gene knockout outcomes, we have included only the result corresponding to the in situ His-tagging of qatA."

The reviewer finds this rationale unconvincing. First, the response does not present any experimental data or evidence supporting the reported knockout efficiencies. Second, it is unclear why these knockout results are considered "redundant," particularly given that demonstrating generality is a central requirement for validating the proposed tool.

4. In the newly added section, the authors state:

"Using the primer set qat_chkF/His-R (Table S2), a PCR product of the expected size (518 bp) was consistently amplified from all 16 randomly selected transformants... No such product was detected in the pAE1 transformant of Δmrr (Fig. 4B)."

However, the reviewer was unable to clearly identify the absence of the PCR product in the pAE1 control in Figure 4B. The figure does not appear to fully correspond to the textual description and therefore requires clarification or additional presentation.

5. The reviewer previously noted:

"In Figure 1, the design rationale of plnt (and pRef) is not sufficiently explained."

The authors responded that:

"A panel has been added to Figure 1D explaining the design rationale of plnt and pRef."

However, in the updated Figure 1D, only the designs of A1S2 and A2S6 are explained, while the design rationale of plnt and pRef is still not explicitly described. Consequently, this concern has not been substantively addressed.

Reviewer #2 (Comments for the Author):

Most of my concerns have been addressed; I do not have other questions.

We sincerely appreciate the reviewers for their insightful comments, which have significantly contributed to improving our manuscript. We have carefully considered all the points raised and have revised the manuscript accordingly. Our point-by-point responses to the reviewers' comments are provided below.

Response to Reviewer 1

Comment 1: *Although the authors have responded to each of my questions, several key major concerns remain unresolved, as detailed below.*

The authors state that: "using the conventional two-step editing approach on the same long deletion targets consistently failed to yield any viable transformants after screening". Therefore, they could not directly compare the two-step approach with the method developed in this manuscript. The reviewer finds this explanation insufficient.

Even if large-fragment editing is challenging for the two-step method, the authors could still perform comparative experiments using shorter fragment lengths to validate one of the claimed advantages of the proposed method, namely reduced time requirements. In addition, the authors should clearly specify from what fragment size onward traditional single- or double-crossover approaches fail to generate clones, thereby defining the practical boundary of applicability for these methods.

At present, the manuscript does not provide a clear or quantitative description addressing these points.

Response: Thank you for the comments. We have previously cited studies highlighting the time-consuming nature of the two-step method in general. Our current work specifically addresses this limitation for *A. faecalis* J481 by developing a CRISPR-based method, as described in INTRODUCTION section:

For example, the classical two-step homologous recombination method has been routinely used for gene knockout in *A. faecalis* (8, 12), which typically involves two steps: replacing the target gene with a selection cassette and subsequently removing the selection cassette through counterselection, generally requiring at least several weeks to obtain a pure mutant strain (13). (lines 61-65) This advancement allowed us to complete two rounds of CRISPR-based genome editing within merely 5 days... (lines 93 & 94)

Although the suggestion to determine the fragment size suitable for generating clones using the traditional “pop-in” method is valuable and will be considered in future research, it falls beyond the scope of this study. Based on our experimental experience, the “largest” genomic segment successfully knocked out using the two-step “pop-in/pop-out” strategy was the 3297-bp *ectABCD* operon in *A. faecalis* J481, a process which we spent approximately six weeks to complete. We anticipate that other researchers may encounter similar challenges, as there are presently exceedingly few reports documenting large-scale gene deletions in this bacterial species.

Comment 2: *In the response, the authors state that the relevant text was revised to:*

"Remarkably, hundreds of transformants were obtained, which is in sharp contrast to the yield of merely 1-2 transformants when hosted by the wild-type J481 cells..."

However, in the revised manuscript, the actual wording is:

"Remarkably, more than 500 transformants were observed, which is in sharp contrast to the yield of merely 1-2 transformants when hosted by the wild-type J481 cells."

No line numbers were provided in the response, making it difficult for me to locate the corresponding revisions in the updated manuscript.

*Moreover, the data indicate that 242 transformants were obtained using pKO-*avs2* prior to *mrr* deletion, whereas 500 transformants were obtained after *mrr* deletion, corresponding to only an approximately twofold increase, which does not appear particularly substantial. The authors argue that the 242 transformants were obtained using 0.5 µg of DNA. It remains unclear whether the comparison described in Lines 194-195 was conducted using equivalent amounts of DNA, and if so, what the exact DNA quantities were. Moreover, 0.5 µg DNA is not an unusually high amount in many bacterial CRISPR-Cas editing experiments.*

Response: We appreciate the reviewer’s comments and apologize for not adequately addressing the concerns raised in the previous round of responses. The amounts of plasmid DNA used in the respective electroporation experiments have now been clearly indicated for efficiency calculation (lines 149, 191 and 198).

Prior to *mrr* deletion, electroporation with 498.03 ng of pKO-*avs2* DNA yielded 242 transformants, corresponding to an efficiency of 485.91 CFU/µg plasmid DNA. We also performed electroporation with 3968.1 ng of pKO-*avs2* DNA under the same conditions, which produced approximately 500 transformants (near the maximum countable range),

reflecting a further reduced efficiency of about 126 CFU/ μ g plasmid DNA. In contrast, after *mrr* deletion, electroporation with 72.24 ng of pKO-*avs2* DNA resulted in over 500 transformants, with a calculated efficiency of about 5.92×10^3 CFU/ μ g plasmid DNA, representing an approximately 15-fold increase compared to the baseline efficiency of 485.91 CFU/ μ g DNA. When 498.03 ng of pKO-*avs2* DNA was used post-*mrr*-deletion, the number of transformants exceeded the countable range. It should be noted that with the small amount (72.24 ng) of plasmid, only 1–2 colonies are typically obtained using the wild-type strain.

For small-gene deletions, it is feasible to use larger amounts of plasmid DNA (e.g., 500 ng or more) to obtain sufficient transformants for subsequent analysis. However, this approach is inadequate for large-fragment deletions. In our experiments aimed at deleting a 30-kb fragment, electroporation with 4,224.93 ng of plasmid DNA yielded only 21 transformants (line 197), with a markedly low transformation efficiency of 4.97 CFU/ μ g plasmid DNA. Thus, the absolute number of transformants does not accurately represent the transformation efficiency. To clarify this, we have explicitly reported the DNA quantity used in each electroporation.

Comment 3: *The reviewer recommended that the authors demonstrate the general applicability of the tool by presenting both knockout and knock-in experiments, and by including at least three different target genes, preferably of different sizes.*

*In their response, the authors state that they achieved: "96% for *qatABCD* operon deletion, 100% for *vap1* deletion, 93.25% for 3PH deletion, and 100% for in situ His-tagging of *qatA*".*

However, the authors further state:

*"to avoid redundant presentation of analogous gene knockout outcomes, we have included only the result corresponding to the in situ His-tagging of *qatA*."*

The reviewer finds this rationale unconvincing. First, the response does not present any experimental data or evidence supporting the reported knockout efficiencies. Second, it is unclear why these knockout results are considered "redundant," particularly given that demonstrating generality is a central requirement for validating the proposed tool.

Response: We sincerely appreciate your valuable comments and fully concur that "demonstrating generality is a central requirement for validating the proposed tool".

Utilizing the optimized methodology developed in this study, we have successfully achieved deletions of several genes in our laboratory during the manuscript submission period. To demonstrate the generality of the developed toolkit, we have included the data showing knockout of genes with representative sizes, including *ectD* (774 bp deleted), *3Hp* (2,634 bp deleted), and *cas3* (3,360 bp deleted), in Figure S2. The results has been describe in the main text as follows (lines 203-208):

Remarkably, using this optimized platform, highly efficient in-frame knockout of genes spanning various sizes was readily achieved, including a 774 bp deletion in *ectD* (*D6I95_05840*) with 100% efficiency (16/16), a 2,634 bp deletion of in the *3Hp* operon (*D6I95_13695-D6I95_13705*) with 93.75% efficiency (15/16), and a 3,360 bp deletion in *cas3* (*D6I95_16065*) with 87.5 efficiency (14/16) (**Fig. S2**). These results substantiate the broad applicability of the native CRISPR-based genome editing toolkit developed for *A. faecalis*.

Fig S2. Efficient knockout of genes of different sizes using the optimized genome editing platform in *A. faecalis*.

(A) Colony PCR screening of deletion mutants of *ectD*, *3PH* operon, and *cas3*. Predicted sizes of PCR products in wild-type (wt) and the intended deletion mutants (Δ *ectD*, Δ *3Ph* or Δ *cas3*) are indicated with unfilled and filled arrowheads, respectively. -, PCR amplification using genomic DNA of *A. faecalis* J481 as a DNA template. M, DNA size marker. **(B)** Representative chromatographs of Sanger sequencing results for the corresponding deletions. Up-flanking (UF) and down-flanking (DF) sequences of each target site are underlined in blue and green, respectively.

Comment 4: *In the newly added section, the authors state:*

"Using the primer set qat_chkF/His-R (Table S2), a PCR product of the expected size (518 bp) was consistently amplified from all 16 randomly selected transformants... No such product was detected in the pAE1 transformant of Δmrr (Fig. 4B)."

However, the reviewer was unable to clearly identify the absence of the PCR product in the pAE1 control in Figure 4B. The figure does not appear to fully correspond to the textual description and therefore requires clarification or additional presentation.

Response: Thank you for your comment. The pAE1 control is marked with "-" in Figure 4B, which has been indicated in the caption (line 588).

Comment 5: *The reviewer previously noted:"In Figure 1, the design rationale of plnt (and pRef) is not sufficiently explained."*

The authors responded that:

"A panel has been added to Figure 1D explaining the design rationale of plnt and pRef." However, in the updated Figure 1D, only the designs of A1S2 and A2S6 are explained, while the design rationale of plnt and pRef is still not explicitly described. Consequently, this concern has not been substantively addressed.

Response: We appreciate your feedback and apologize for not explicitly articulating the design rationale of plnt and pRef. We have revised Figure 1, using a plasmid map to illustrate the key genetic elements and the protospacer insertion site. The design is described in Figure 1 caption as follows (lines 559-563):

(D) Plasmid interference assay. Interference plasmids (plnt) contain a protospacer comprising a spacer preceded by a 5'-CCC-3' PAM, while the corresponding reference plasmids (pRef) feature a 5'-AAA-3' motif substituted for the 5'-CCC-3' PAM. *EcoRI* and *BamHI* restriction sites used for protospacer insertion are indicated. Nucleotide sequences of the protospacers for A1S1 and S2S6 are presented. Two interference plasmids, plnt-A1S1 and plnt-A2S6, were used to assay DNA interference activity of the Type I-F CRISPR-Cas. Transformation efficiencies of each interference plasmid are expressed as relative values to the efficiencies of their corresponding reference plasmids (pRef-A1S1 and pRef-A2S6), the latter of which are set to 1.0. Experiments were done in triplicate. Error bars represent the standard deviation of the mean.

Response to Reviewer 2

Comment: *Most of my concerns have been addressed; I do not have other questions.*

Response: We deeply appreciate your time and efforts in assisting us to improve our manuscript.

Re: Spectrum02786-25R2 (**Unlocking genome engineering in *Alcaligenes faecalis* by exploiting its native Type I-F CRISPR-Cas**)

Dear Dr. Wenfang Peng:

Your manuscript has been accepted, and I am forwarding it to the ASM production staff for publication. Your paper will first be checked to make sure all elements meet the technical requirements. ASM staff will contact you if anything needs to be revised before copyediting and production can begin. Otherwise, you will be notified when your proofs are ready to be viewed.

Sincerely,
Jeffrey Gardner
Editor
Microbiology Spectrum